# WWFedCBMIR: World-Wide Federated Content-Based Medical Image Retrieval

**DOI:** 10.3390/bioengineering10101144

**Published:** 2023-09-28

**Authors:** Zahra Tabatabaei, Yuandou Wang, Adrián Colomer, Javier Oliver Moll, Zhiming Zhao, Valery Naranjo

**Affiliations:** 1Department of Artificial Intelligence, Tyris Tech S.L., 46021 Valencia, Spain; 2Instituto Universitario de Investigación en Tecnología Centrada en el Ser Humano, HUMAN-Tech, Universitat Politècnica de València, 46021 Valencia, Spain; 3Multiscale Networked Systems, Universiteit van Amsterdam, 1098XH Amsterdam, The Netherlands; 4ValgrAI—Valencian Graduate School and Research Network for Artificial Intelligence, 46022 Valencia, Spain

**Keywords:** breast cancer, content-based medical image retrieval (CBMIR), convolutional auto-encoder (CAE), federated learning (FL), computer-aided diagnosis, histopathological images, digital pathology, whole-slide images (WSIs)

## Abstract

The paper proposes a federated content-based medical image retrieval (FedCBMIR) tool that utilizes federated learning (FL) to address the challenges of acquiring a diverse medical data set for training CBMIR models. CBMIR is a tool to find the most similar cases in the data set to assist pathologists. Training such a tool necessitates a pool of whole-slide images (WSIs) to train the feature extractor (FE) to extract an optimal embedding vector. The strict regulations surrounding data sharing in hospitals makes it difficult to collect a rich data set. FedCBMIR distributes an unsupervised FE to collaborative centers for training without sharing the data set, resulting in shorter training times and higher performance. FedCBMIR was evaluated by mimicking two experiments, including two clients with two different breast cancer data sets, namely BreaKHis and Camelyon17 (CAM17), and four clients with the BreaKHis data set at four different magnifications. FedCBMIR increases the F1 score (F1S) of each client from 96% to 98.1% in CAM17 and from 95% to 98.4% in BreaKHis, with 11.44 fewer hours in training time. FedCBMIR provides 98%, 96%, 94%, and 97% F1S in the BreaKHis experiment with a generalized model and accomplishes this in 25.53 fewer hours of training.

## 1. Introduction

Breast cancer accounts for 25% of all cancers in women worldwide. According to the American Cancer Society, a woman is diagnosed with breast cancer in the world every 14 s. In the year 2020, approximately 2.3 million women were diagnosed with breast cancer globally, and 685,000 lost their lives due to it [1]. Histopathology is commonly used in the diagnosis and treatment of various diseases, including cancer. A biopsy, which is the removal of a small piece of tissue from the body, is usually required for histopathological examination [2]. Human error in histopathology refers to mistakes or inaccuracies made during the process of examining tissues or cells under a microscope [3]. Some examples of human errors in histopathology include sampling errors, processing errors, technical errors, interpretation errors, and reporting errors [4]. To minimize human errors in histopathology, it is essential to follow strict protocols and guidelines, perform regular quality control checks, and ensure that all personnel involved in the process are properly trained and competent [5]. The authors in [6] analyzed the accuracy of breast cancer diagnosis in 102 cases and found that there were diagnostic errors in 15.7% of cases. The most common types of errors were misclassification of tumor type and misinterpretation of pathology slides. Digital pathology could help pathologists to improve the accuracy and efficiency of cancer diagnosis, reduce the risk of errors, and enhance patient care.

Digital pathology is a technology that uses digital images of tissues and cells to aid in the diagnosis and management of diseases [7]. Deep learning (DL) has revolutionized computer-aided diagnosis (CAD) in digital pathology and has opened the door to improve cancer diagnosis while decreasing the pathologist’s workload [8].

Content-based medical image retrieval (CBMIR) is a recent DL-based methodology that allows pathologists a quick and precise search in previously diagnosed and treated cases [9]. In CBMIR, image features, such as texture, shape, color, and intensity, are extracted from the query and data set; then, a similarity measure is applied to compare the query features with the features of the database [10]. The retrieved images are ranked according to their similarity to the query image, and the most relevant images are displayed to the user.

To further illustrate the advantages and practicality of CBMIR in the field of histopathology and cancer diagnosis, consider a scenario where a patient is diagnosed with cancer, and grading it accurately poses a challenge for pathologists. In traditional cancer diagnosis methods, the pathologist would need to physically send the glass slide containing the tissue sample to another hospital, which could be located in a different city or even a different country. This process is not only expensive and time-consuming but also carries inherent risks, such as the loss or damage of the glass slide during transportation. Moreover, it adds additional stress to the patient’s already difficult situation.

By implementing world-wide content-based medical image retrieval (WWCBMIR), these challenges can be effectively addressed, and the process of a cancer diagnosis can be significantly expedited without compromising accuracy. Through the use of digital pathology, where whole-slide images (WSIs) are digitized and stored electronically, pathologists can access and analyze the images remotely [2]. The WWCBMIR enables pathologists to retrieve similar cases and relevant information from a vast database of histopathological images without the need for the physical transfer of slides. This approach not only reduces costs and saves time but also minimizes the potential risks associated with the transportation of delicate tissue samples. Figure 1 shows how a WWCBMIR can provide unprecedented access to *K* number of patches with the most similar patterns, allowing the pathologists to make a more confident diagnosis.

One of the advantages of CBMIR from the pathologist’s (user) perspective is that it is not a completely black box for them. CBMIR allows pathologists to find similar patterns among the retrieved images and the queries based on their knowledge. This provides more reliable information than a label for pathologists, which makes CBMIR more beneficial for pathologists than a classification.

An actual context needs a global CBMIR, which demands a generalized data set with a variety of images of different quality, magnification, color, size, etc. The performance of CBMIR relies on a vast amount of data, which is difficult to collect in the medical field due to patient privacy and time costs. In order to create a vast centralized data set, DL experts need to transfer their WSIs. However, these images are gigapixels with high storage sizes. In addition to the challenges of transferring a heavy data set for DL experts, patient privacy policies and other regulatory obstacles on the medical side make it more challenging to create a sufficient data set.

Federated learning (FL) represents a possible solution to tackle this problem by collaboratively training DL models without transferring WSIs [11]. Multiple institutions can safely co-train DL models in digital pathology using FL, achieving cutting-edge performance with privacy assurances [12]. FL brings an opportunity to share the weights for multi-institutional training without sharing patient data and images. However, there are still some privacy risks since the training parameters and model weights are distributed among collaborators [13].

DL models give information that goes beyond the scope of human vision, and FL solves the problem of data sparsity by connecting international hospitals while complying with the data privacy policies, irrespective of the country of origin. This benefit can remedy the health care limitations due to the lack of facilities (staining materials, scanners, etc.) and experience (students, recently graduated pathologists, etc.). Moreover, it can tackle the lack of data sets of labeled WSIs because of data privacy.

In this paper, we minimized the WWCBMIR to an international CBMIR by leveraging FL. The experiments were conducted through the collaboration of two countries and three cities to examine the feasibility and challenges associated with implementing a WWCBMIR. This international CBMIR was trained with the data collected from different hospitals and answered the needs of clients. Clients might be expert pathologists or a student. Our main contributions include the following:We proposed a novel international FL-based CBMIR, which is named FedCBMIR, to aid pathologists in breast cancer diagnosis.An unsupervised network was used as a feature extractor (FE) to extract the features of the images for the tasks trained with scanty data sets.We proposed a custom-built convolutional auto-encoder (CAE) to learn the dependencies and extract the features of the images with higher discriminating values.In order to address patient data privacy concerns, we employed the privacy preservation capability of FL. This approach ensures that the data in each institution remains decentralized and confidential, as there is no need to be shared with a central server.Through extensive tests on varying data set distributions among individual clients, we verified the robustness of our proposed solution. It proved to be independent of the data quality held by each client.

## 2. Related Work

Recently, researchers have directed their attention toward both FL and CBMIR and have invested their efforts in exploring these fields. This section provides a succinct overview of some of the notable studies.

### 2.1. Content-Based Medical Image Retrieval (CBMIR)

CBMIR has been a subject of extensive research since the advent of large-scale databases nearly two decades ago, as noted by Wang [14]. Several studies have made significant contributions to this field. Tabatabaei [15] achieved an accuracy rate of 84% in CBMIR using the largest patch-annotated data set in prostate cancer. Kalra [16] proposed Yottixel, a method for representing the Cancer Genome Atlas whole-slide images (TCGA WSIs) compactly to facilitate millions of high-accuracy searches with low storage requirements in real time. Conversely, Mehta [17] proposed a CBMIR system for sub-images in high-resolution digital pathology images, utilizing scale-invariant feature extraction. Lowe [18] utilized scale-invariant feature transform (SIFT) to index sub-images and reported an 80% accuracy rate for the top-five retrieved images. Lowe’s experiments were conducted on 50 ImmunohHistoChemistry (IHC) stained pathology images at eight different resolutions. Additionally, Hegde [19] used a manually annotated data set pre-trained on a deep neural network (DNN) to achieve top-five scores for patch-based CBMIR at different magnification levels. The primary focus of recent studies has been on enhancing the performance of CBMIR in different types of cancer; however, there are still several challenges that can impede its effectiveness. These challenges include data privacy, as medical data is confidential and subject to strict privacy regulations, making it arduous to share and access large data sets for model training. FL can alleviate this issue by facilitating distributed model training on local data without compromising privacy. Another challenge is data distribution; as medical data is frequently dispersed across numerous locations, it is difficult to train models on a centralized data set. FL enables the training of models across multiple distributed data sets without aggregating the data in a central location. In addition, medical data sets can be heterogeneous, varying in terms of imaging modalities, quality, and annotation protocols, which can impede the development of robust and accurate models. FL can mitigate this challenge by allowing models to be trained on diverse data sets in different qualities, improving their performance and generalization ability. Furthermore, medical data sets can be large and complex, necessitating significant computational resources to train models. FL can distribute the computational workload across multiple devices and locations, enhancing scalability and reducing training time.

### 2.2. Federated Learning (FL)

In recent years, FL has achieved impressive progress that enhances a wide adoption of DL from decentralized data [11,20,21]. FL is a distributed machine learning approach that can effectively handle decentralized data without raw data exchange to train a joint model by aggregating and distributing local training. Many existing algorithms can be adopted to aggregate updates from distributed clients. Typical examples include FederatedAveraging, *viz* FedAvg [11], and adaptive federated optimization methods [21], e.g., FedAdagrad, FedYogi, and FedAdam. Some popular FL frameworks, such as TensorFlow Federated (TFF) (https://www.tensorflow.org/federated (accessed on 23 September 2022)), PySyft [22], and Flower [23] provide a set of robust tools for building privacy-preserving ML models. In addition, Jupyter-Notebook-based tools, such as [24], also help simplify the FL setup and enable its deployment of a cross-country federated environment in only a few minutes. Daniel Truhn in [25] employed homomorphic encryption to protect the model’s performance while training by encrypting the weight updates before sharing them with the central server. Firas Khader in [26] presented a technique of “learnable synergy”, where the model only chooses pertinent interactions between data modalities and maintains an“internal memory” of key information. Micah J. Sheller [13] investigated how FL among ten institutions is 99% as efficient as that derived using centralized data. One recent work related to content-based image retrieval is introduced in [27], where FLSIR was proposed, and it enables secure image retrieval based on FL and additive secret sharing. Nevertheless, it is not for clinical applications. Although the combination of CBMIR and FL is a relatively new area of research, it has the potential to greatly improve healthcare outcomes. By offering healthcare professionals quick access to accurate and relevant medical image data while maintaining patient privacy, the integration of these techniques can have a significant impact on the field.

The following sections address how the proposed FedCBMIR approach can revolutionize how medical images are searched and utilized, leading to improved diagnoses and treatment plans.

## 3. Experiments

In this section, the proposed FedCBMIR tool is introduced along with the training details and the two data sets used in our study. Figure 2 (BreaKHis images) provides an overview of the CBMIR workflow, starting from the initial stage at a hospital and concluding with the presentation of the top *K* similar patches to the user. In the medical session, a cancer patient’s tissue is obtained, scanned, and divided into patches for storage. In the offline session, the FE is trained, and the extracted features from the database are saved and indexed. In the online session, a pathologist uploads an image to the CBMIR model, where the well-trained FE extracts its features. These features are then used by the search engine to retrieve the top *K* similar patches from the stored database in the medical session. Finally, in the visual session, the pathologist can reach similar patches and their corresponding labels for further investigation based on their knowledge.

In this paper, the proposed FedCBMIR, as shown in Algorithm 1 (more information: https://flower.dev/docs/framework/how-to-implement-strategies.html (accessed on 23 September 2022)), addresses the described challenges and provides a second opinion for pathologists in writing their reports for a cancer diagnosis. FedCBMIR is inspired by a great vision of a WWCBMIR that effectively manages decentralized medical images by utilizing local training for multiple tasks while avoiding the need for raw data exchange. FedCBMIR takes advantage of FL since it can give CBMIR a higher chance of generalizing its capabilities by accessing multi-central images from different hospitals. A generalized CBMIR framework needs more effective content of the images as the key factor in the field of CBMIR.

In this paper, we cope with the challenges of CBMIR with two different experiments and evaluate it in three scenarios. In our first experiment (EXP 1), we mimic a case of two institutions that have different breast cancer WSIs in completely different image preparation processes. This case occurs when two institutions have a limited number of images, but they need a well-trained model to obtain a supportive idea on their query tissue. This experiment was assessed this experiment on CAMELYON17 (CAM17) and BreaKHis at 40× magnification. Then, in the second experiment (EXP 2), we extended our work with patches at different magnifications by feeding our FedCBMIR framework with BreakHis data set at 40×, 100×, 200×, and 400× magnification. The magnification problem in WSI analysis is the subject of our second experiment. Algorithm 1 shows FedCBMIR step by step. The novelty of this work relies on providing well-trained models that can retrieve similar patches for each client in different countries. Regarding the use of FL in CBMIR, all clients, regardless of their data privacy policies, can train the model with a limited number of patches and find similar patches to their queries more accurately than local training.
**Algorithm 1** FedCBMIR(FedAvg)
  **Server (Aggregator)**                  **Client (CBMIR)**
  *Initialization*………………………………………………………………………………………………………………………
  *M ◃ The number of clients*
  *R ◃ The communication rounds*
  *E ◃ The local epochs*
  *B ◃ The local batch size*                *H               ◃ hyperparameters*
  η* ◃ The local learning rate*               M*            ◃ model structure*
  $\omega_{0}$*◃ weights*                     $D_{m}$*            ◃ local data set of client*
                             *m*
  *Phase 1*……………………………………………………………………………………………………………………………1: **for all** round r=1,2,…,R **do**2:  $S_{r}=\left(\mathrm{random}\;\mathrm{set}\;\mathrm{of}\;M\;\mathrm{clients}\right)$3:  **for all** client $m\in S_{r}$ **do**4:   $\omega_{r+1}^{m}=\mathrm{ClientUpdate}\left(m,\omega_{r}\right)$                              **ClientUpdate:**     *◃ execute on client m*
                             5:  train $D_{m}$ with model M structure                             6:
    $\beta \leftarrow (\mathrm{split}\;P_{m}\;\mathrm{into}\;\mathrm{batches}\;\mathrm{of}\;\mathrm{size}\;B)$                             7:  **for all** localepochi=1,2,…,E **do**                             8:    **for all** batchb∈β **do**                             9:      ω=ω−η∇ι(ω;b)                               **return** ω to server  *Phase 2*……………………………………………………………………………………………………………………………  **FederatedAveraging:** *◃ execute on server*10:   $\omega_{r+1}=\sum_{m=1}^{M}\frac{n_{m}}{n}\omega_{r+1}^{m}$


The performance of FedCBMIR was validated on histopathological images using a CAE in a cross-institutional distributed environment. FL was used as a collaborative learning paradigm in which the CAE can be trained across different institutions without explicitly sharing data sets.

### 3.1. Materials

Hematoxylin and eosin (H&E) is a type of histopathological staining. H&E has been popular for almost a century because it may indicate morphological changes [28]. The images in the data sets used in this paper were stained by H&E.

#### 3.1.1. BreaKHis

BreaKHis contains 7909 histopathological images of breast tumor tissues that were provided by a collaboration with the P&D Laboratory—Pathological Anatomy and Cytopathology, Parana, Brazil. This data set was collected from 82 patients at four magnifications (40×, 100×, 200×, and 400×) with 2480 benign and 5429 malignant cases. As can be understood in Table 1, the number of images in benign and malignant cases is imbalanced. The most considerable portion of the data set belongs to the images at 100× magnification (https://www.kaggle.com/datasets/ambarish/breakhis (accessed on 23 September 2022)).

#### 3.1.2. CAMELYON17 (CAM17)

The CAM17 data set belonging to the CAMELYON17 challenge, as described by [29], is designed to detect breast cancer metastasis in lymph node sections. It comprises 1000 WSIs obtained from five distinct hospitals. Each hospital contributed data from 20 patients, with five slides per patient, and annotations for cancer regions were provided for a subset of 50 WSIs. In this paper, images from four hospitals were used for training and validating the model, and the images of Hospital 5 were fed into the model as a test set. Non-overlapping 224 × 224 (at 40×) pixel patches with at least 70% tissue were used for experiments on this data set. In the experiments of this paper, the data set was considered as a binary data set, including Cancerous (annotated) and Non-Cancerous (not annotated) images.

### 3.2. Data Distribution

The CLoud ARtificial Intelligence For pathologY (CLARIFY) project (http://www.clarify-project.eu/ (accessed on 23 September 2022)) has a multi-institutional paradigm. In this work, according to the connections between different institutions in CLARIFY, four institutions (three universities and one company) in three cities in two countries gathered to mimic the practical situation of FL in CBMIR.

In EXP 1, in order to distribute the data into two nodes, we assume that Tyris (TY) (Spain, Valencia) and the Universiteit van Amsterdam (UvA) (Amsterdam, The Netherlands) have CAM17 and BreaKHis 40×, respectively. As can be seen in Table 2, TY caries out training the FedCBMIR on a GPU resource in the type of NVIDIA GeForce RTX 3090. The GPU used in UvA is the NVIDIA Tesla T4, which has fewer CUDA cores, slower memory clock speed, and lower memory bandwidth compared to the GPU used in TY. These different GPUs are chosen to mimic the real condition of different hospitals or research centers having different GPU performance.

In EXP 2, regarding mimicking the real-world data limitation, the four magnifications of the data set were distributed into four nodes. To accomplish this, each institution (client) in this paper has BreakHis at only one magnification to train their model (Table 3). Universidad de Granada (UGR) (Spain, Granada), TY, UvA, and Universidad Politécnica de Valencia (UPV) (Spain, Valencia) trained the custom-built CAE with BreakHis 40×, 100×, 200×, and 400×, respectively. To replicate real-world conditions where clients may not have access to high-performance GPUs, our experiment includes three distinct GPU types across four institutions. This ensures alignment with practical scenarios and provides a comprehensive evaluation of different GPU capabilities.

### 3.3. Training the Convolutional Auto-Encoder in Each Node

One of the most crucial elements of CBMIR that influences search engine results is the FE. The objective of content-based image search is to efficiently compare an extracted feature from a query image to every image in a database to identify the matches that are most similar.

Lack of annotated images and bias are the two major challenges that need to be considered in the integration of DL into cancer diagnosis. Three factors have the potential to cause bias in medical studies: data-driven, algorithmic, and human bias. To tackle these obstacles, a custom-built CAE is configured as the FE in this paper as a generative model where it is trained to reconstruct its input in an unsupervised way. The proposed structure of CAE contains a skip layer to jump over the layers to not only lead the model to converge faster and minimize the training errors but also boost the representation power and tackle the vanishing problem. Also, it has a residual block in its bottleneck to enable the training of deeper and more accurate CAE.

Figure 3 shows its architecture with convolutional filters in the size of [32,64,128,256] in the encoder and, respectively, [128,64,32,3] in the decoder. In this custom-built CAE, a residual block with the filter size of [64,32,1,256] takes place between the encoder and decoder. This takes the originally extracted features from the backbone as its input and provides a new feature map that contains the context relations between its feature input. In our experiments, a skip layer connect a layer in the encoder to the corresponding layer in the decoder. The bottleneck delivers one feature vector with 200 features ($F_{i}=\left\{f_{1},f_{2},f_{3},\ldots ,f_{200}\right\}$) from each encoded input image *i*. The model aims to achieve the lowest mean squared error (MSE) by comparing input (*I*) and output (*O*) and is penalized if the reconstruction *O* differs from *I*. Once the unsupervised training is completed by discarding the decoder part, a powerful automatic FE is available to extract the desired features.

### 3.4. Local Training

Figure 4 (BreaKHis images are used to plot the figure.) explains the whole pipeline of the proposed CBMIR that each institution must follow to retrieve similar patches. In the offline session, images in the training and validation set are fed into the FE to extract and save their features as in the previous cases. All the $F_{i}$s are collected in a dictionary $D=[F_{1},F_{2},\ldots ,F_{n}]$ in the middle of this figure.

In the online session, pathologists upload their patch as a query image (*Q*) and expect to receive top *K* similar patches. In practice, each *Q* needs to feed to the FE and map to its feature vector $F_{Q}$. Then, $F_{Q}$ feeds to the distance metrics in order to compare with the $F_{i}$s saved in *D*. To accomplish this, in our experiments, as soon as the pathologists upload their *Q*, the *Q* image is fed to the FE to extract $F_{Q}$ with 200 features. Then, the Euclidean function applies on both $F_{Q}$ and the $F_{i}$s in *D* to measure their similarity and deliver top *K* similar patches.

### 3.5. Federated Learning Configuration

In order to train the CBMIR following a federated strategy, different experiments have been conducted on FedAvg and FedAdagrad. In our cases, with some experiments, it is found that FedAvg performs better than FedAdagrad. Thus, this work adopts FedAvg to aggregate distributed updates from local clients, as shown in Algorithm 1:(1)$$\omega_{r+1}=\sum_{m=1}^{M}\frac{n_{m}}{n}\omega_{r+1}^{m}$$
where *M* indicates the number of clients, *r* presents the communication round. For a client *m* with $n_{m}$ samples, the local updates are arbitrary: $\omega_{r+1}^{m}$.

FLOWER [23], as a primary framework, is applied to configure the FL experiments. Two FL experiments were conducted, as shown in Figure 5a,b. The first experiment consists of two distributed training nodes located in TY and UvA (see Figure 5a). In the two communication rounds, the learning rate is set as 0.000001, 5 local epochs for CAM17 per round, and 100 local epochs for BreakHis 40×. Also, FedCBMIR is extended with more clients, as shown in Figure 5b. The system consists of four separate nodes, each of which is trained using the BreaKHis data set at different magnifications. The training process involves three communication rounds and a learning rate of 0.000001, and each client performs 100 local epochs per round. Table 3 lists all four distributed processing nodes’ information in the training phase.

## 4. Discussion and Results

### 4.1. Evaluation

To allow for an adequate comparison of the model’s performance, three metrics were selected: accuracy (ACC), precision, and F1 score (F1S), in addition to presenting the confusion matrix (CM). Accuracy assesses how well a model correctly retrieved similar patches to the query [30]. Precision measures the accuracy of positive predictions, which is vital when false positives are costly. The F1S combines precision and recall into a single metric [31]. In this paper, to evaluate the proposed FedCBMIR, each of the images in the test set was considered a query. Across the entire training and validation set, the model is meant to detect similar patches.

It is worth considering what “accuracy” means in the context of a CBMIR. The accuracy of CBMIR depends on what we are looking for and what is displayed by the search engine. In order to determine the performance of the experiments, the top *K* score of retrieving images of the same histologic features are engaged from prior research. The evaluation method will consider a correct answer from the model whenever it finds at least one correct image within the *K* set [15]. In this paper, we set K=5, which evaluates the performance of our model to correctly present at least one correct result in the top *K* retrieved images.
(2)$$ACC@K=\frac{1}{N}\sum_{i}^{N}\varepsilon \left(\alpha_{i},TOP\left(ans\left[:K\right]\right)\right)$$

In this equation, *N* denotes the number of query patches, and $\alpha_{i}$ represents the label of the *i*-th query patch. The function $TOP(ans_{i}\left[:K\right])$ retrieves the top *k* most similar results for the query and outputs 1 if any of these results match with the query, and 0 otherwise. In other words, if $TOP(ans_{i}\left[:K\right])$ belongs to the set of labels of the *i*-th query, denoted by $\alpha_{i}$, the function ε() returns 1.

### 4.2. Results of EXP 1

For this particular experiment, BreaKHis 40× and CAM17 data sets were aggregated to train the model. As a result, each client (UvA and TY) could develop a well-trained model to retrieve their respective images. The underlying assumption made in this experiment is that neither client had an agreement in place for sharing or accessing each other’s images. Table 4 provides a comprehensive view of the model. As it is mentioned above, CAM17 was provided by five hospitals. To perform this evaluation, the CAM17 images from Hospital 5 were isolated from the images in the other four hospitals that were utilized for the training and validating task. Each image from Hospital 5 serves as a query in the testing assignment, and the platform’s function is to seek patches with a similar pattern from the other four hospitals. Table 4 illustrates that the accuracy of local training of CAM17 without aggregating with BreaKHis is less than the FedCBMIR with aggregated data. This table indicates that FedCBMIR using the FedAvg approach achieved better results than FedCBMIR using FedAdagrad. As a result, FedAvg was selected as the aggregation technique for the subsequent experiments.

In terms of time and accuracy, local training of the CBMIR model on BreaKHis40× and CAM17 requires 9.33 and 8.7 training hours, resulting in an accuracy of 93% and 96% in the test set, respectively. However, FedCBMIR was trained more efficiently and achieved a higher accuracy level of 98.1% in retrieving similar patches in CAM17, and 97.8% accuracy for UvA, with a reduction of 2.49 and 2.74 h in training time, respectively. In order to have two distinct models on both data sets separately, 18.04 h are needed, while FedCBMIR trains two generalized models on both data sets in 6.59 h (Max(6.21 h, 6.59 h)=6.59 h). This means FedCBMIR provides more generalized models for clients 11.44 h faster.

Training time and accuracy are essential factors for DL scientists in building an optimal model, whereas accuracy and searching time are crucial for pathologists in retrieving similar patches. The table shows that the TY client can obtain a second opinion with labels and similar patches in only 0.28 s per image. Upon examining the “Training time” and “Searching time” columns, it becomes evident that the utilization of FL has no noticeable impact on the searching time, while it substantially influences reducing the training time.

Figure 6 represents three random queries in the test set of CAM17 with their top-five retrieved images among the training and validation sets. Figure 7 represents the comparison of image search results with two CMs in the test set of CAM17 as a result of local training (CBMIR) and FedCBMIR.

### 4.3. Results of EXP 2

In EXP 2, the performance evaluation of the proposed framework was conducted using two distinct scenarios. The first scenario, ***Sen1***, assumed that the clients did not have access to images from other clients, and it was only allowed to share the model weights during the training phase. This scenario was designed to test the performance of the framework when the participating clients faced technical limitations in sharing large amounts of medical imaging data. In this scenario, each client had to train their model on their local data, and the models’ weights were shared with other clients. Then, the weights were combined and trained using the entire data set from all participating clients. Finally, the model was evaluated on each client’s local test set.

***Sen1*** mirrors the situation where clients can only obtain patches that are similar to their *Q* at the same magnification. Because there is no explicit agreement among the institutions, the model is obliged to search for similar cases in a few cases at that particular magnification.

Table 5 summarizes the results of the proposed FedCBMIR on the BreaKHis data set at all four magnifications. This table shows the accuracy and precision of the retrieved images at each magnification, achieved by each client after training their models for 300 epochs within their server and without using FL. The highest accuracy of 95% for the retrieved images at 40× magnification was achieved by the client at UGR in 9.37 h, while client 3 spent 8.59 h to achieve a minimum accuracy of 89% and precision of 87%, which is the lowest among all the clients.

As demonstrated in Table 5, using the proposed approach, the four models were trained in a federated setting, which took (Max(6.82 h, 5.78 h, 6.65 h, 6.83 h)=6.83 h) hours to complete the training process, which is much faster than training one by one that took 32.36 h in total, thereby reducing the total training time around 25.53 h. This reduction in training time is particularly significant for large data sets and can facilitate more rapid and accurate diagnoses and treatments of cancers.

The performance evaluation of the proposed framework in the test set was compared with local training CBMIR and FedCBMIR, as shown in Figure 8. Each CM is associated with a specific magnification and reports the top-five accuracy values using ***Sen1*** in its search stage. The results of the ***Sen1*** in the test set are presented in Figure 8, where each client receives the top-five images on average in 13.84 s. Table 6 presents a comprehensive comparison of various state-of-the-art CBMIR methods on the BreaKHis data set at 40× magnification. It is evident that FedCBMIR achieved the highest performance in both experiments conducted in this paper (EXP 1 and EXP 2). Notably, in EXP 1, where the model was trained by sharing weights with the CAM17 client, FedCBMIR exhibited superior performance. Previous studies by [32,33] utilized a hash method on BreaKHis 40× images and reported results in 16-bit, 32-bit, and 64-bit formats. Since the best-reported performance was achieved with the 64-bit format, we compare our results solely with this format, excluding 16-bit and 32-bit comparisons.

Quantifying mitosis count is a crucial criterion in breast cancer diagnosis [34]. The availability of advanced technology, such as high-resolution scanners, is not always guaranteed in every part of the world. Figure 9 demonstrates that as the magnification level increases, a smaller area of the tissue is displayed, and more relevant information becomes visible.

Figure 10 shows a comparison between the proposed method and its obtained results under EXP 1 and EXP 2 *Sen1* condition. The methods mentioned in the figure were applied to a binary breast tissue microscopic image data set built in [32,35]. In paper [33], 20 retrieved images were taken into consideration in evaluating their proposed method. In Figure 10, since the authors in [33] did not name their two methods, we named them Method1 and Method2, then compared their results with the top 20 retrieved images with our results at the top 5 images. As can be understood from the bar charts in Figure 10, FedCBMIR in both experiments could overtake the other methods, and they have higher precision in the retrieval performance. It is important to mention that Method1 and Method2 in [33] are supervised and reached 87% and 89.5% precision by proposing a supervised hashing method with multiple features, while FedCBMIR with an unsupervised FE obtained precision equal to 97% and 96% for both EXP 1 and EXP 2, *Sen1*.

**Table 6 bioengineering-10-01144-t006:** A comparison is presented between the average precision values of state-of-the-art papers and the results obtained in the reported experiments of this paper (EXP 1 and EXP 2, *Sen1*).

Methods	CBMIR, EXP 1	FedCBMIR, EXP 1	FedCBMIR, EXP 2, *Sen1*	Method [36]	MCCH [37]	KSH, 64 Bits [32]	JKSH, 64 Bits [33]
**Precision**	0.93	**0.97**	0.96	0.95	0.94	0.91	0.87

The proposed ***Sen2*** approach can serve as an important tool for pathologists in developing nations to overcome the limitations of their scanners by enabling them to access tissue images at higher magnifications. FedCBMIR can facilitate cross-border collaborations, where pathologists from different regions can share their knowledge and expertise by analyzing similar patches at higher magnifications. In contrast to CBMIR, FedCBMIR in ***Sen2*** allows pathologists to retrieve similar cases at all four magnifications, not just from the same magnification as their query (*Q*). However, sharing images with a single server is not feasible due to storage and privacy concerns. To address this issue, the proposed FedCBMIR can retrieve similar patches at the same and higher magnifications.

Table 7 proves that the proposed FedCBMIR is highly robust to receiving a query at a specific magnification and retrieving the top-five similar patches at all four magnifications. Each client fed the test set at the corresponding magnification and received the top-five retrieved patches at all four magnifications.

The results of feeding the model with five random queries at 40× magnification by following (***Sen2***) are presented in Figure 11. The 40× is selected because it is the lowest magnification in the data set, and it is easier to measure the number of mitoses in images with higher magnifications. By feeding the model with images at 40×, pathologists can receive the top-five similar images at 40×, 100×, 200×, and 400×, which can significantly reduce the time and effort required to obtain a second opinion. The proposed approach has the potential to improve the speed and accuracy of cancer diagnosis and treatment. As such, it can serve as a user-friendly platform for pathologists to address their concerns more. Furthermore, it has the potential to be a valuable tool for telepathology in the future.

One of the challenges in collecting WSIs for use in DL models is the variability in color distribution due to differences in the staining material used across different hospitals and over time [38]. This variability can have a significant impact on the accuracy and reliability of DL models. However, an important finding from the results shown in Figure 11 is that the proposed approach, ***Sen2***, is not affected by differences in color distribution resulting from the staining process at different hospitals. This is a noteworthy result, as it indicates that the proposed approach can effectively overcome one of the major challenges associated with collecting and utilizing WSIs in telepathology. By eliminating the impact of color distribution variability, ***Sen2*** provides a more robust and reliable platform for pathologists to obtain accurate and consistent diagnoses, regardless of the specific staining materials used at different centers.

The proposed approach can contribute significantly to improving the accuracy and speed of disease diagnosis, particularly in regions where access to advanced technology is limited. In this way, ***Sen2*** has the potential to bridge the gap in healthcare and provide a more equitable and accessible healthcare system for all.

All the experimental results in both experiments and scenarios have verified that the proposed FedCBMIR has covered both the concerns of DL scientists and pathologists with a fast-trained and accurate CBMIR, which is more generalized.

## 5. Conclusions

The present study proposes a FedCBMIR approach that addresses two significant challenges in digital pathology faced by pathologists and engineers. By retrieving the top-five similar images in a short amount of time, the proposed method reduces the workload of pathologists and decreases the time and cost associated with developing a high-performing DL-based method.

To evaluate the proposed approach, two experiments (EXP 1 and EXP 2) were conducted, with EXP 2 containing two scenarios. EXP 1 aimed to provide a generalized model with Camelyon17 (CAM17) and BreaKHis 40× for clients that do not have enough images to train a model effectively. FedCBMIR in EXP 1 provides precision of 97.0% and 96.9% for training an unsupervised feature extractor 11.44 h faster.

EXP 2 comprised two scenarios: *Sen1*, where image institutions are not in agreement for sharing images, and *Sen2*, where images are delivered in different magnifications for institutions that lack the equipment to scan tissues at higher magnifications. The proposed method reached 98%, 96%, 94%, and 97% F1S for each client in *Sen1*. In *Sen2*, the BreaKHis data set was distributed across four institutions, resulting in accuracy rates of 97%, 94%, 92%, and 96% for pathologists at magnifications of 40×, 100×, 200×, and 400×, respectively. The average retrieval time was 13.84 s, and the well-trained models required 25.53 fewer hours to train the four generalized models.

On one hand, WWCBMIR provides a chance to have a more accurate diagnosis for less-developed countries.

On the other hand, FedCBMIR can be a valuable tool for new graduate pathologists in their training and professional practice, offering benefits such as improved education, decision-making, research, and time efficiency.

Overall, this work offers a promising tool for hospitals to enhance diagnostic accuracy and medical education and reduce the workload of pathologists by decreasing training time and increasing accuracy compared to CBMIR methods. FedCBMIR aids in recognizing rare cases by connecting hospitals from the whole of the world. Although FedCBMIR tackles the challenges of data privacy, limited clinical context, and algorithm accuracy, the ongoing issues, such as dependence on image quality and security concerns, are still challenging for both hospitals and AI experts. Therefore, both hospitals and engineers must weigh the advantages and drawbacks while considering WWFedCBMIR as a tool.

## 6. Future Work

To further enhance the performance of FedCBMIR for breast cancer diagnosis, it may be worthwhile to explore the use of additional data sets. This could include larger data sets with a greater number of labeled images, as well as data sets that encompass a wider range of malignancy levels and tumor subtypes. With the incorporation of these data sets into the FL process, it may be possible to improve the accuracy and robustness of a CBMIR.

In addition to expanding the data sets used in federated CBMIR, it may also be valuable to incorporate other types of clinical data into the system. Patient demographic information and clinical history could provide additional context and help to further refine the diagnostic process. Exploring the integration of these types of data could be a promising avenue for future research.

## Figures and Tables

**Figure 1 bioengineering-10-01144-f001:**
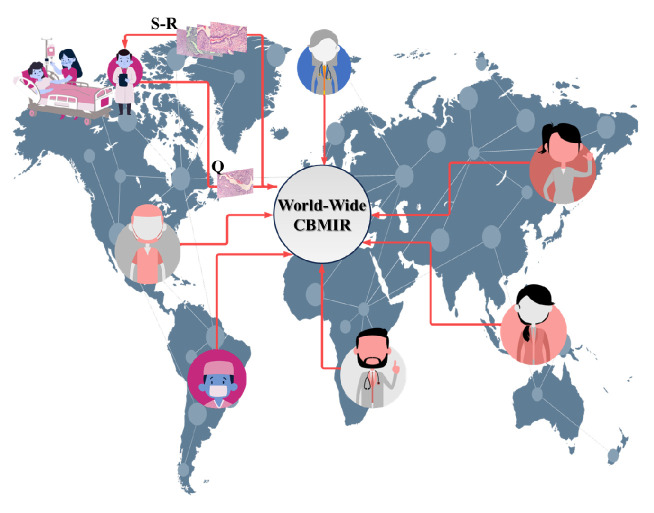
An overview of the use case of a worldwide CBMIR. Pathologists send their query (**Q**) to the worldwide CBMIR since they need a second opinion to make a more confident decision. Then, the model retrieved top *K* similar images (**S-R**), and the pathologists can obtain a second opinion from whole over the world.

**Figure 2 bioengineering-10-01144-f002:**
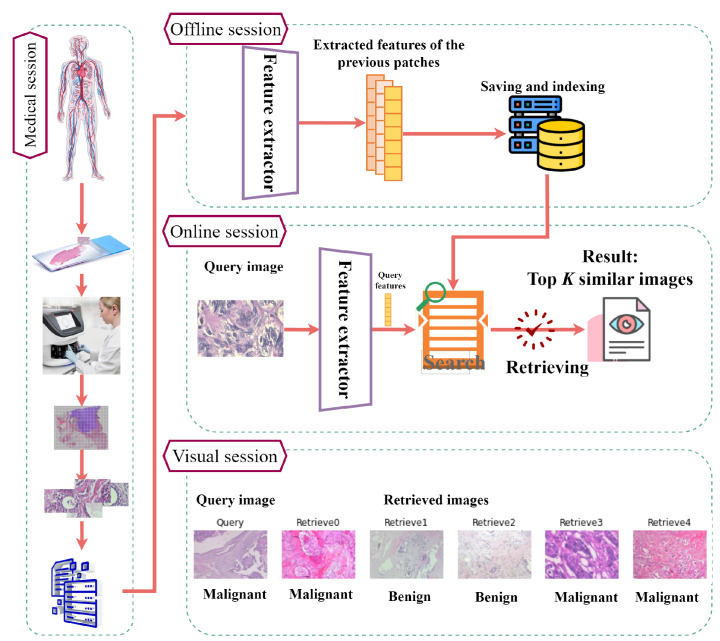
A comprehensive illustration of the entire process in a CBMIR, demonstrating the utilization of DL models to acquire images from a hospital and offer a second opinion for pathologists.

**Figure 3 bioengineering-10-01144-f003:**
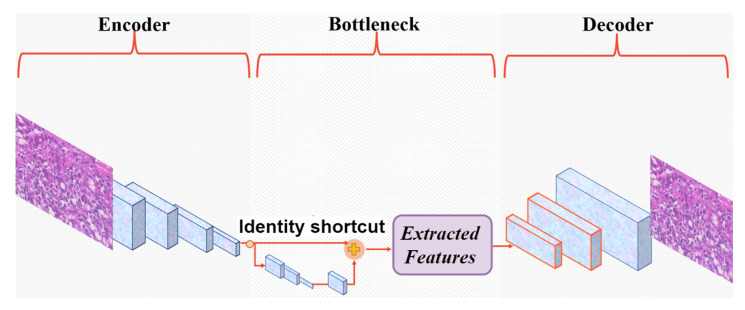
The structure of the custom-built CAE. The stride in the encoder = [1,2,2,2], in the bottleneck = [1,1,1,1], in the decoder related to the encoder = [2,2,2,1]. The kernel size of the layers in all parts of the structure and for each layer is 3.

**Figure 4 bioengineering-10-01144-f004:**
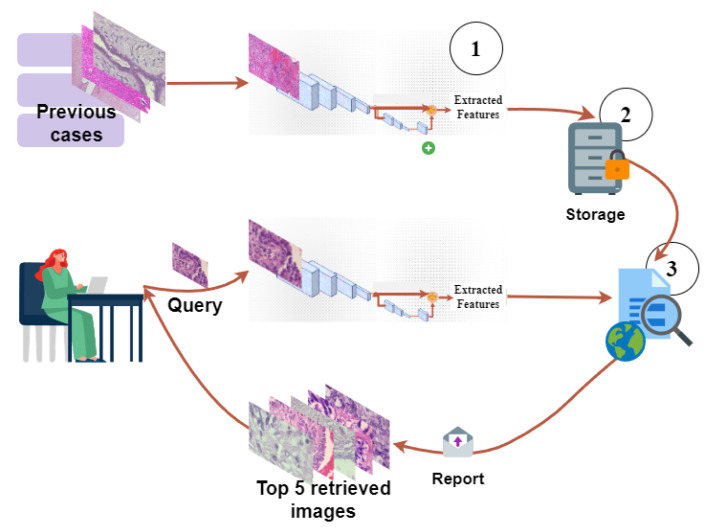
The pipeline of CBMIR. It contains three important sections, namely (1) FE, (2) indexing and saving, and (3) similarity measure and search.

**Figure 5 bioengineering-10-01144-f005:**
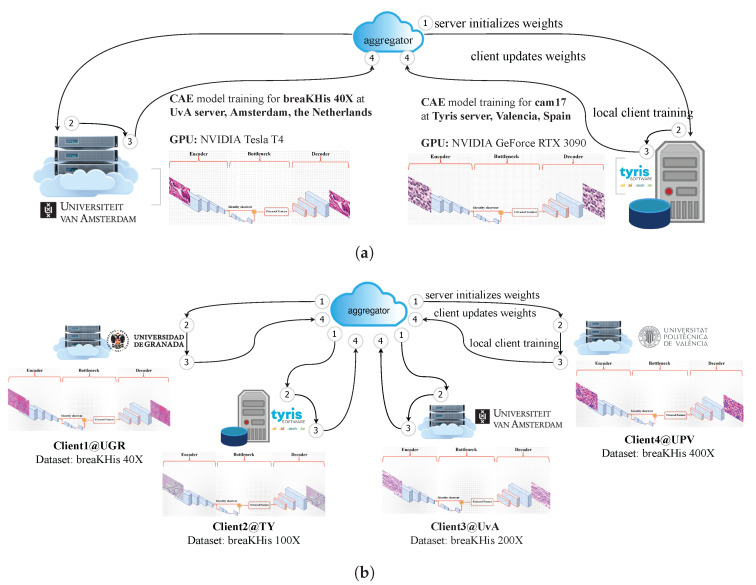
The FedCBMIR pipeline consists of four main steps. Step 1: the server initializes weights, and then sends to client for local training. Step 2: client starts local training. Step 3: client updates local weights to the server side. Step 4: the server side aggregates and updates the distributed weights. (**a**) An overview of the FedCBMIR pipeline with two clients training, fed with BreaKHis 40× and CAM17 data sets. (**b**) An overview of the FedCBMIR pipeline with four clients training over clusters at universities and companies with BreaKHis in four different magnifications.

**Figure 6 bioengineering-10-01144-f006:**
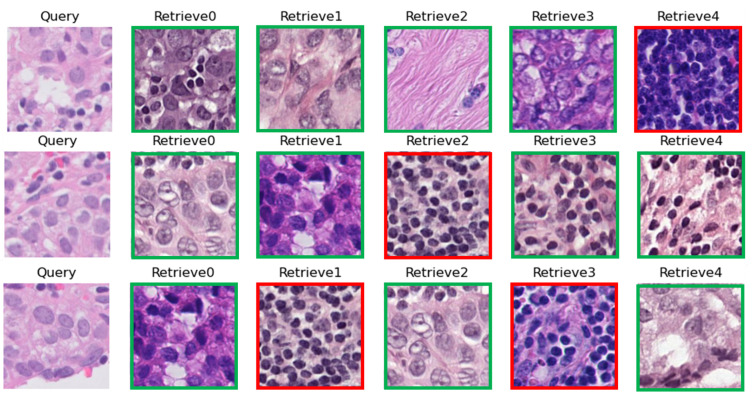
Three random queries from Hospital 5 of CAM17 (test set). Corresponding to each query, the top 5 images are shown from four other hospitals with the most similar patterns to the query. The green and red lines around the retrieved images explain the correct and wrong retrieved images.

**Figure 7 bioengineering-10-01144-f007:**
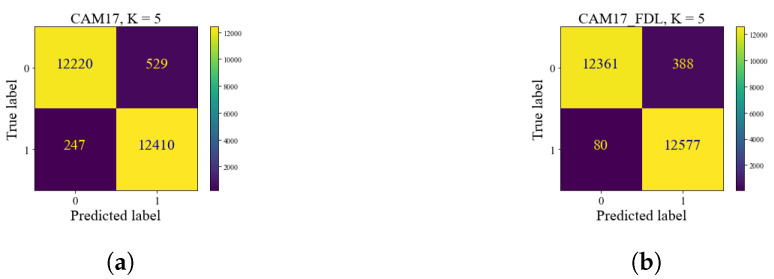
(**a**) shows the results of local training on CAM17 in the TY server. (**b**) is the result of the searching task in CAM17 by applying the well-trained FedCBMIR model from the first experiment.

**Figure 8 bioengineering-10-01144-f008:**
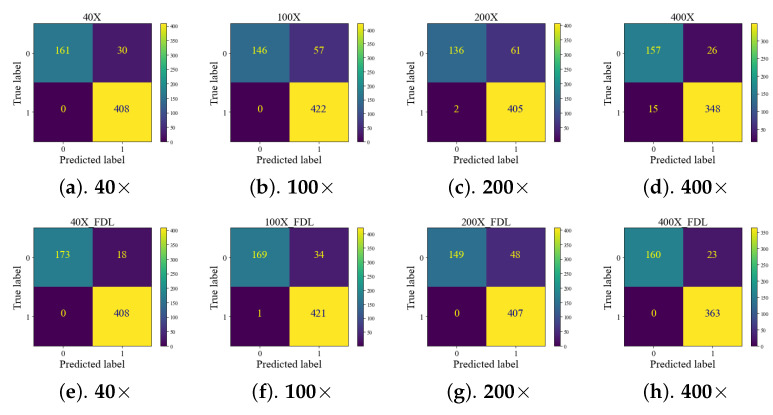
(**a**–**d**) show the CMs as a result of local training and searching at the same magnification. (**e**–**h**) are the CMs of FL models. The reported results are with top *K* retrieved images. In all CMs, "0” and “1” indicate “**Benign**” and “**Malignant**”, respectively. “**True labels**” and “**Predicted labels**” correspond to the query and the retrieved labels, accordingly.

**Figure 9 bioengineering-10-01144-f009:**
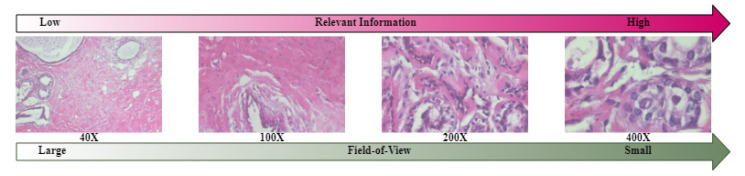
BreaKHis images at four different magnification levels (40×, 100×, 200×, and 400×). The higher magnification offers increased access to relevant information with a reduced field of view.

**Figure 10 bioengineering-10-01144-f010:**
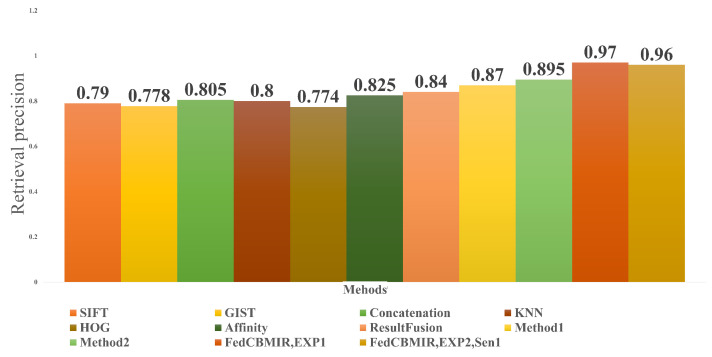
An indirect comparison between the results of FedCBMIR in both experiments and some recent methods for different amounts of *K*.

**Figure 11 bioengineering-10-01144-f011:**
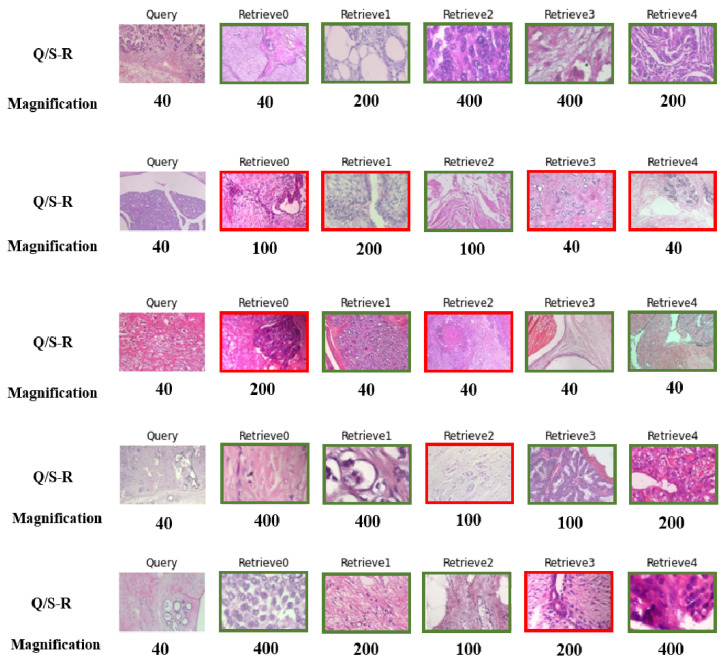
Five lines of random histopathological WSIs with their magnifications. The first column is the query, and the following five columns show the retrieved images. This figure brings a proper overview of ***Sen2***. The retrieved image with the same and different labels as the query is indicated by the green and red borders, accordingly.

**Table 1 bioengineering-10-01144-t001:** The distribution of BreakHis data set.

Magnification	Benign	Malignant	Total
40×	625	1370	1995
100×	644	1437	2081
200×	623	1390	2013
400×	588	1232	1820
Total	2480	5429	7909

**Table 2 bioengineering-10-01144-t002:** Data distribution in EXP 1. The information of each institution participating in EXP 1, including their location, the name of their center, the data associated with their data distribution, and the GPUs employed by each client for training and searching tasks.

Client	Region	Institution	Data Set	GPU Type
1	Valencia, Spain	TY	CAM17	NVIDIA GeForce RTX 3090
2	Amsterdam, The Netherlands	UvA	BreakHis 40×	NVIDIA Tesla T4

**Table 3 bioengineering-10-01144-t003:** Collected information on each client in EXP 2, including their country and city, the name of the center, the related data due to the data distribution, and the GPUs used for training and search tasks by each client.

Client	Region	Institution	Magnification	GPU Type
1	Granada, Spain	UGR	40×	NVIDIA GeForce RTX 3090
2	Valencia, Spain	TY	100×	NVIDIA GeForce RTX 3090
3	Amsterdam, The Netherlands	UvA	200×	NVIDIA Tesla T4
4	Valencia, Spain	UPV	400×	NVIDIA TITAN V

**Table 4 bioengineering-10-01144-t004:** Comparison of the test set between the performance of CBMIR and FedCBMIR in EXP 1 as a result of aggregating CAM17 and BreaKHis 40× with 2 communication rounds. Hours and seconds, respectively, are used to measure the periods of training and searching.

Data	Model	Accuracy	Precision	F1S	Training Time	Searching Time
**CAM17** **(TY)**	CBMIR	0.96	0.96	0.96	8.7 h	0.28 S
	FedCBMIR (Fedavg)	**0.981**	**0.970**	**0.981**	**6.21** h	0.29 S
	FedCBMIR (FedAdagrad)	0.98	0.97	0.98	7.92 h	0.30 S
**BreaKHis** **40×** **(UvA)**	CBMIR	0.93	0.94	0.95	9.33 h	0.018 S
	FedCBMIR (Fedavg)	**0.978**	**0.969**	**0.984**	6.59 h	0.024 S
	FedCBMIR (FedAdagrad)	0.94	0.92	0.96	**6.11** h	0.04 S

**Table 5 bioengineering-10-01144-t005:** Obtained results of CBMIR on 40×, 100×, 200×, and 400× at K=5. We measure ACC, Precision, and F1S in the test set of each client at their corresponding magnification. The FedCBMIR was trained with the FedAvg strategy with 5 communication rounds in EXP 1. Time is reported in hours.

Client	Model	Training Time	Accuracy	Precision	F1S
1	CBMIR	9.37 h	0.95	0.93	0.96
	FedCBMIR	6.82 h	**0.97**	**0.96**	**0.98**
2	CBMIR	5.45 h	0.90	0.88	0.94
	FedCBMIR	5.78 h	**0.94**	**0.92**	**0.96**
3	CBMIR	8.59 h	0.89	0.87	0.93
	FedCBMIR	6.65 h	**0.92**	**0.89**	**0.94**
4	CBMIR	8.95 h	0.92	0.89	0.94
	FedCBMIR	6.83 h	**0.96**	**0.94**	**0.97**

**Table 7 bioengineering-10-01144-t007:** The ACC, precision, and F1S for the second scenario of the EXP 2, *Sen2* with *K* = 5.

Client	Accuracy	Precision	F1S
1	0.94	0.92	0.95
2	0.95	0.93	0.96
3	0.95	0.93	0.96
4	0.95	0.92	0.96

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
