# Peer review of "WWFedCBMIR: World-Wide Federated Content-Based Medical Image Retrieval"

_bioengineering, 2023, doi:10.3390/bioengineering10101144_

Round 1
Reviewer 1 Report
In this manuscript, the authors proposed a novel FL-based CBMIR to improve the accuracy and efficiency of breast cancer diagnosis. The manuscript addressed the difficulty of creating a sufficient dataset, proposed an unsupervised model, and utilized federated learning to train this model, obtaining greater accuracy with less training time. The manuscript created experimental scenarios to simulate real-world hospital configurations, data privacy, and data distribution issues. The experimental results showed that the proposed approach outperforms other approaches. This study holds significant interest for readers. However, some clarifications are necessary before publication.
1. Section 3 provides a comprehensive overview of the current benefits and challenges associated with Content-Based Medical Image Retrieval (CBMIR), as well as outlines the potential application scenarios for the Federated CBMIR (FedCBMIR) system. This is quite similar to Section 1. Other scholars proposed the FedAvg, and it is recommended that a detailed description of the algorithm be included in the appendix. I suggest that Sections 1 and 3 should be combined and shortened.
2. The title of Table 2 is incorrect. I believe that the name is “Collects information on each client in EXP 1”. Isn't that so?
3. In Subsection 4.3, the custom-built CAE is the proposed network, What is the difference between a custom-built CAE and the traditional CAE, the author needs to provide more details.
4. In Subsection 5.1, a brief description of accuracy, precision, and F1-score would help readers better grasp the evaluation procedure.
5. In the conclusion, discuss and highlight the benefits that can be provided to the hospital. Additionally, the limitations or potential drawbacks of the proposed technique should be discussed.
Minor editing of the English language required.
Author Response
Dear contact person
Please see the attachment
Thanks
Best
Zahra

Reviewer 2 Report
Reviewer report
The scientific Article from Zahra Tabatabaei, Yuandou Wang, Adrián Colomer, Javier Oliver Moll, Zhimming Zhao, and Valery Naranjo, with the title:
WWFedCBMIR: World-Wide Federated Content-Based Medical Image Retrieval
The article describes a method to train a CBMIR model via a distributed FL training, thereby omitted the need for acquiring and sharing large datasets, which is both demanding and includes privacy issues.
The article first describes the basic concepts of the method, and how the method would be implemented on a worldwide basis. To show the implementation of the method the author’s proceeds with a reduced form of the method called FedCBMIR, which is tested on two different datasets breast cancer containing images of either BreaKHis or CAMELYON17. The training is shown on different GPU equipment, and images at different magnifications.
The results from the test of the FedCBMIR method show lower training time and higher accuracy than the standard CBMIR method.
In my opinion, the scientific field Federated Content-Based Medical Image Retrieval is well suited for the bioengineering journal and the presented work is of high standard and could be published after minor revision. I would like the authors to consider the following points before recommending publication:
To improve the current manuscript:
1) The English text is of high standard and very pleasant to read. I however found the in the discussion and results a few places:
a. Line 346 and 348 “the” should be removed in front of hospital 5.
b. Line 416 “but” should be change to “and”
2) The figures and tables are relevant to the manuscript and are general of high quality. The authors should consider the following:
a. Figure 1: The text is small and hard to read
b. Figure 3: The text in the is small and hard to read
c. Figure 6: In the caption change “5th hospital” to “Hospital 5” to keep consistency with the text
d. Figure 7: In caption change “7a” and “7b” to “(a)” and “(b)” to consistent. The legends on the figure is small and hard to read. The “s” on the label “Actuals” should be removed.
e. Figure 8: The legends on the figure is small and hard to read. The “s” on the label “Actuals” should be removed.
f. Figure 9: In the caption “offering” should be changed to “offer an”
g. Table 5: The last sentence in the caption “Time is reported with an hour”, should this be: Time is reported in hours?
3) Line 281, [128, 64, 32, 3], should the 3 be an 8?
4) Line 402, what bar chart is referred to?
5) In table 6: Method [ref 34] and MCCH [ref 35] are mentioned in the text
The reviewer have the following questions to the manuscript:
1. Regarding the 200 features extracted for each image, Line 288. In the training of the model, are all features equally important? And can the user of the proposed model chose to emphasis a subset of the 200 features?
2. In the proposed Sen2 with different magnifications, in Line 423 and forward, is there a problem if there is for instance is 4 images at different magnification from the same patient. Could the model return these 4 images out of the 5 total returned images, as a consequence of removing the privacy issues? Or is this taken care of in the data preparation?
3. The model is shown on breast cancer images, and the reviewer therefore assumes all patients are female. But for other cancer types, and if the model was scaled to the WWFedCBMIR model, would such things as gender and ethnicity have to be taken into account for in the comparison?

The English is of high standard, and only minor errors detected.
Author Response
Dear contact person
Please see the attachment.
Best
Zahra
